# Predictive effects of diabetes-related risk factors for falls in community-dwelling people with diabetic peripheral neuropathy based on a logistic regression model

Eneida Yuri Suda[1,2,3], Cristina Dallemole Sartor[1], Anice de Campos Passaro[1], Ricky Watari[1], Eunice Young Docko[1], Isabel C. N. Sacco[1]*

1 Physical Therapy, Speech and Occupational Therapy Department, School of Medicine, University of São Paulo, São Paulo, Brazil, 2 Master Program in Physical Therapy, Universidade Ibirapuera, São Paulo, Brazil, 3 Masters and Doctoral Programs in Physical Therapy, Department of Physical Therapy, Universidade Cidade de São Paulo, São Paulo, Brazil

* icnsacco@usp.br

## Abstract

### Background

This study aimed to identify the predictive effects of different aspects of diabetic peripheral neuropathy (DPN) and other already known risk factors for falls through a comprehensive logistic model within community-dwelling older adults with diabetes and DPN. This paper also provides a model that estimates the probability of a fall occurring in a real-world clinical scenario.

### Methods

This cross-sectional retrospective study analyzed data from subjects that had never fallen (non-fallers, n = 534) and that had fallen at least twice in the previous year (fallers, n = 101). The logistic regression analysis was performed on a training sample randomly extracted from the original sample (non-fallers: n = 85; fallers: n = 81). The model was validated by checking the performance parameters using a test sample comprised of 10% of fallers (n = 16) and a proportionate subsample of non-fallers (n = 85) from the original dataset.

### Results

Three predictive models were developed. The best model (0.762 receiver operating characteristic[ROC] curve area, 60.4% accuracy, 68.8% sensitivity, 58.8% specificity) identified age (odds ratio[OR]=1.06[95%CI: 1.02, 1.10], P = 0.002), Michigan Neuropathy Screening Instrument score (OR=1.23[95%CI: 1.08, 1.40], P = 0.001), and self-reported balance problems (OR=2.65[95%CI: 1.29, 5.45], P = 0.008) as predictors of falls. A second model with good performance parameters (0.750 ROC

**Data availability statement:** The datasets generated and/or analyzed during the current study are accessible as anonymized data at https://doi.org/10.6084/m9.figshare.30546620.

**Funding:** This work was supported by the National Council for Scientific and Technological Development, Brazil (CNPq) [28/2018 FOCA Trial 407252/2018-5]. Sacco is a fellow level 1B of the National Council for Scientific and Technological Development (CNPq), Brazil (Process: 302558/2022-5) and Docko holded a scholarship from CNPq (PIBIC). All funders did not have any role in the study and do not have any role in the study and do not have any authority over any study activity or in the decision to submit the report for publication. There was no additional external funding received for this study.

**Competing interests:** NO authors have competing interests.

curve area, 62.4% accuracy, 62.5% sensitivity, 62.4% specificity) showed that age (OR=1.04[95%CI: 1.01, 1.07], $P$ = 0.015), balance problems (OR=3.29[95%CI: 1.64, 6.59], $P$ = 0.001), and DPN severity (OR=1.18[95%CI: 1.03, 1.34], $P$ = 0.018) were predictors of falls.

## Conclusions

We showed the potential of a predictive model for recurrent falls based on commonly evaluated variables in community-dwelling individuals with diabetes for use in clinical practice. Even for individuals who are not at a high risk for falls, it is crucial to assess the combination of DPN signs, symptoms, and severity and the perception of balance problems, as these are more relevant in people with diabetes than the traditional physical impairments associated to aging.

## Introduction

Falls are among the top 10 causes of loss of disability-adjusted life years in individuals ages 65 years and older [1]. Approximately 25% of older adults between 65 and 74 years old fall every year, and these rates increase with further aging [2]. Falls are a multidimensional phenomenon and, as such, their multiple factors must be addressed [3]. Within the biological dimension, the well-known fall-related factors are aging, reduced visual quality, stroke, urinary incontinence, cognitive impairments, depression [4], hypotension [5], obesity, dizziness and vertigo [6], polypharmacy, alcohol abuse [7], fear of falling, and pain [8]. Furthermore, alterations in blood glucose levels, such as hypoglycemia [5] and sustained hyperglycemia [6], have been identified as risk factors for falls, and they are closely linked to diabetes mellitus (DM), a condition that is also independently associated with higher chances of suffering falls [9–11].

Falls are particularly more prevalent in more severe DM patients, especially in people with retinopathy, nephropathy, vestibulopathy, autonomic neuropathy, and diabetic peripheral neuropathy (DPN) [4,5,12,13]. Notably, DPN severely affects sensorial perception under the feet and foot-ankle proprioception [14,15], which are crucial for postural and gait control [16]. These afferent disturbances in DPN also result in deviations in joint kinematics during gait [17–20], which impacts the foot clearance in the swing phase [21] and increases the chances of stumbling and falling [9–11]. However, because falling is a multidimensional phenomenon, it can be challenging to determine the independent contributions of several musculoskeletal, biomechanical, and neurological deficits related to DPN to the occurrence of a fall [22]. Thus, studying the independence and interplay of different risk factors for falling among people with DM and DPN may be an important step in fall prevention strategies and rehabilitation planning.

Logistic regression models are a powerful way of identifying the independence and interplay of specific risk factors for falls in individuals with DM and DPN while estimating the probability of an event's occurrence in this population [23]. For the past 20

years, models have been built to identify risk factors in this population, but they all lack an informative description of the performance parameters of the model for further use as a tool for estimating falls occurrence or do not include a comprehensive set of DPN-related variables for better characterization of DPN severity degree.

In the first model reported, Schwartz et al. [24] concluded that falling more than once a year was associated with a decrease in vibratory sensitivity and loss of tactile sensitivity in a prospective study; however, the logistic model was not described in a way that it could be applied to the general population, since only the odds ratios (ORs) of the adjusted logistic regression models were reported but not the coefficients for each variable. In addition, the performance parameters of the model were also not reported. MacGilchrist et al. [25] identified walking velocity, ankle dorsiflexors strength, and neuropathy symptom score, as independent risk factors for falling; nevertheless, their study presented a small sample with imbalanced groups (fewer fallers than non-fallers), and once again only the OR for each predictor was described. Shah et al. [26] showed that severe hypoglycemia, depression, and DPN were associated with an increased fall risk in people with type 1 DM; however, once again, the authors only reported the OR for each predictor. The authors also did not disclose how DPN was assessed, since the information was extracted from medical charts. Rinkel et al. [27] found that tactile sensory loss, assessed by the 39-item Rotterdam Diabetic Foot Study Test Battery, was a significant predictor of recurrent falls, but the study only took into account tactile perception deficit as a factor related to DPN, missing all other sensorial (vibratory) and motor (range of motion and strength) components.

Although the aforementioned studies showed that different variables could increase the risk of falling in this population and presented evidence that DPN is a solid risk factor for falls, to date, not a single model has presented a prediction structure that takes into account the interplay between different aspects of DPN and other already known risk factors for falls, which could also be used as indicators in clinical practice. Thus, the aim of this study was to identify predictive risk factors for falls within community-dwelling older adults with DM and DPN, taking into account the DPN severity degree, and to provide a predictive comprehensive logistic model that allows for the estimation of the probability of a fall occurring in a real-world clinical scenario.

## Methods

### Study design and measures

This cross-sectional retrospective study was based on the database of the 2014 and 2015 Annual Brazilian Campaign of Diabetes Detection, Orientation, Education and Complication Prevention by the National Association of Diabetes Assistance in São Paulo, Brazil. This study was approved by the Ethics Committee on Research of the University of Sao Paulo Medical School (protocol number 054/10), and all subjects signed an informed consent form.

### Participants

A total of 765 subjects with DM type 1 or 2, of both sexes (360 men, 351 women; mean age = 64 ± 10 years old), who had participated in the 2014 (November 9th, 2014) and 2015 (November 9th, 2015) Annual Brazilian Campaign of Diabetes Detection, Orientation, Education and Complication Prevention were included in the analysis. Participants who were wheelchair dependent, with amputations above the toe level, with prosthesis on the lower limbs, or pregnant were not included. Users of gait-assistive devices were included in this study.

### Procedures

**DPN signs and symptoms and severity assessment.** The evaluations were performed by 15 physical therapists who were rigorously trained to guarantee reliability between assessments and were closely supervised during the procedures. DPN signs and symptoms were assessed using the Brazilian version of the Michigan Neuropathy Screening Instrument (MNSI) [28]. Scores ranged from 0 to 13, with a higher score indicating worse symptoms. DPN severity was

determined using the Decision Support System for Classification of Diabetic Polyneuropathy (www.usp.br/labimph/fuzzy) [29,30]. This fuzzy expert system includes as input variables the vibration perception, tactile sensitivity, and DPN symptoms assessment. The system performs a combinatory analysis of the input variables using an *if-then* rule base, linking them with fuzzy output sets that are transformed into a numerical value by the center of area defuzzification method, resulting in a DPN degree score that ranges from 0 to 10 points. Higher scores indicate more severe DPN. The fuzzy expert system was tested and showed high sensitivity and specificity in discriminating between patients with and without DPN (receiver operating characteristic [ROC] = 0.985).

A 10-g monofilament (Sorri-Bauru®, Bauru, SP, Brazil) was used to randomly assess the tactile sensitivity of four plantar areas (first, third, and fifth metatarsal heads and on the distal phalanx of the hallux) following the instructions described by Bakker et al. [31]. Individuals with open wounds or ulcers were not tested. For the vibratory sensitivity, a 128-Hz tuning fork was used on the bone prominence of the interphalangeal hallux joint [31,32].

**Falls and related conditions.** Patients were asked about the occurrence of falls in the previous 12 months as well as potential causes and conditions when the fall happened (stood up too fast, stumbled, lost balance, was pushed, fainted, lost strength in the legs, and other causes). Subjects who had fallen at least twice in the previous year were considered fallers in this study. Furthermore, individuals self-assessed their balance as good or bad, whether they felt weakness in their limbs, and whether they feared falling. We also inquired about the presence of a visual deficiency and the number of medications taken.

**Functional reach test.** Functional balance was assessed using the functional reach test proposed by Duncan et al. [33]. The participants stood barefoot, feet together and upper limbs elevated at 90 degrees of flexion parallel to the ground. A measuring tape was placed on the wall, parallel to the ground. The participants were instructed to lean forward as much as possible without elevating the ankles, losing balance, flexing the hips, or taking a step. Displacement was checked on the measuring tape. The difference between the position of the extremity of the third finger at the beginning and end of the test was calculated. The greater the distance achieved, the better the functional balance. This tool is considered an efficient predictor of falls in older adults when the reaching distance is less than 25.4 cm (established as a normative value and as the minimum threshold for a low risk of falls) [33,34]. A distance between 15.2 and 25.4 cm indicates a falls risk twice as high as the first range, and for individuals who reach distances less than 15.2 cm, the risk is four times greater than the first group [33]. Subjects who presented a functional reach test result of less than 15 cm were classified as at risk of falling in this study.

## Statistical analysis

The following variables were selected for the logistic regression analysis (Table 1): age, sex, visual deficiency, medication intake, risk of falling based on the functional reach test [6], use of a gait aid, self-reported balance problems, self-reported limbs weakness, self-reported fear of falling, time since DM diagnosis, fasting glucose level, signs and symptoms of DPN, tactile and vibratory sensitivities, and DPN severity degree. Since multiple falls are associated with an increased risk of falling, while isolated falls are not, the predicted variable was determined as the occurrence of two or more falls [27]. From the original sample (N = 765), 534 participants had never fallen (non-fallers), 130 had fallen once (not included in the analysis), and 101 had fallen two or more times (fallers). The logistic regression equation was determined using a training sample dataset. The training sample consisted of a subsample of 90% of the fallers from the original sample (n = 81) and an equivalent subsample of non-fallers (n = 85) to avoid misclassifications of the model due to the imbalanced occurrence of the fallers and non-fallers in the original sample. This subsample was randomly extracted from the original sample.

Initially, simple logistic regression analyses were performed for each of the independent variables (Table 1) to determine which variables would be included in the final model. An alpha level of 0.2 was determined for this selection to avoid the exclusion of potential confounding variables. Then, the pre-selected variables were checked for multicollinearity through a multiple regression analysis. When two variables showed high collinearity (correlation coefficient > 0.7), only one

**Table 1. Description of the variables considered for the logistic regression.**

| Variable | Score range/ Categorization |
| --- | --- |
| Age (years) | > 18 years old |
| Sex | 0 = male; 1 = female |
| Fasting glucose level (mg/dL) | > 70 |
| Visual deficiency | 0 = no; 1 = yes |
| Use of gait aids | 0 = no; 1 = yes |
| MNSI total score | 0-13 |
| Polypharmacy (use > 4 medications) | 0 = no; 1 = yes |
| Balance (self-reported) | 0 = good; 1 = bad |
| Weakness in the limbs (self-reported) | 0 = no; 1 = yes |
| Fear of falling (self-reported) | 0 = no; 1 = yes |
| Tactile sensitivity (# insensitive plantar areas) | 0-8 |
| Vibration sensitivity | 0 = present; 1 = absent |
| Falls risk (Functional Reach Test) | 0 = low (<15 cm); 1 = high (≥15 cm) |
| DPN severity (fuzzy score) | 0-10 |

MNSI: Michigan Neuropathy Screening Instrument; DPN: diabetic peripheral neuropathy.

was chosen to be included in the model. The logistic regression was performed with the selected variables for the training sample in both a forward and backward stepwise fashion. A significance level of 0.10 was set for the inclusion and/or removal of the variables for the steps of both methods. The significance of the coefficients was determined by means of likelihood ratio tests, and the overall goodness of fit of the model was tested with the Hosmer–Lemeshow test. The variables included in the final model were used to generate the logistic regression equation 1 (S1 File). The ability of the model to discriminate between non-fallers and fallers was assessed by the ROC curve. Since the logistic regression was performed in both a forward and backward stepwise fashion, the ROC curve results were used to select the model with better performance.

The model was validated by checking the performance parameters using a test sample consisting of the remaining 10% of fallers from the original sample (n = 16) and a subsample of non-fallers (n = 85). The number of non-fallers included for the validation was determined to maintain the proportion of the cases in the original sample. These subjects were extracted randomly from the original sample and were not used to build the model. The logistic regression equation generated by the model was applied to each of the test set observations, and the predicted variable of falls was estimated according to Equation 2 (S1 File). The performance of the predictive model on the test set was assessed for accuracy, precision, recall, Matthews correlation coefficient (MCC), and ROC curve. To determine which magnitude of a determined variable would predict that an individual would become a faller, the ROC curves were calculated for the predicted probability obtained for each case by the logistic model, and the coordinate points with a sensitivity of 0.8 were used to determine the cut-off.

Independent sample t-tests were used to compare the numerical variables between the non-fallers and the fallers to determine the logistic model. Chi-square tests were used to compare the frequency of occurrence of the nominal variables between non-fallers and fallers. All statistical analyses were performed using SPSS Statistics 24 (IBM, Armonk, NY, USA), with statistical significance defined as $p < 0.05$.

## Results

Participants in the fallers group showed on average three falls in the previous year, were significantly older, were less likely to be male, and were more likely to have a visual deficiency (Table 2). Fallers also presented with more DPN

**Table 2. Anthropometric, demographic, and clinical variables (mean±SD) for the studied groups included in the model.**

| | Non-fallers (n = 85) | Fallers (n = 82) | *P* |
|---|---|---|---|
| Age (years) | 61.2±11.0 | 65.5±9.9 | 0.017†* |
| Sex (% male) | 58.8 | 37.8 | 0.007‡* |
| Fasting glucose level (mg/dL) | 178.6±79.7 | 174.3±86.3 | 0.741† |
| Visual deficiency (%) | 27.1 | 42.7 | 0.034‡* |
| Use of gait aid (%) | 8.2 | 7.3 | 0.825‡ |
| Polypharmacy (%) | 43.5 | 48.8 | 0.496‡ |
| Number of falls in the past 12 months | 0.0±0.0 | 3.0±1.6 | <0.001†* |
| MNSI total score | 2.8±2.6 | 4.5±3.0 | <0.001†* |
| Tactile sensitivity (# insensitive plantar areas) | 1.2±2.1 | 1.9±2.3 | 0.027†* |
| Vibration sensitivity (% of absent sensation) | 7.1 | 14.6 | 0.115‡* |
| DPN severity (Fuzzy score) | 2.6±2.4 | 3.4±2.7 | 0.001†* |
| Self-reported balance problems (%) | 24.7 | 53.7 | <0.001‡* |
| Self-reported limbs weakness (%) | 29.4 | 47.6 | 0.016‡* |
| Self-reported fear of falling (%) | 40.0 | 65.9 | 0.001‡* |
| High risk of falling (FRT<15cm) (%) | 11.8 | 22.4 | 0,034‡* |

MNSI: Michigan Neuropathy Screening Instrument, FRT = functional reach test.

*Statistical difference, † Independent t-test, ‡ X² test.

symptoms (higher MNSI scores); a higher number of foot areas with tactile sensitivity loss; higher number of subjects with vibratory sensitivity loss and more severe DPN degree (fuzzy score); worse balance (lower functional reach test scores); and higher number of individuals with self-reported balance problems, fear of falling, and limb weakness.

The simple logistic regression analysis for each of the independent variables identified 11 potential risk factors with a significance level of at least 0.02 (S1 Table), with fasting glucose level, use of a gait aid, and polypharmacy being excluded from the final logistic models. The collinearity analysis showed that the DPN severity presented high correlation with vibration and tactile sensitivities, which are inputs in the Fuzzy system that determine DPN severity. Therefore, we chose to calculate two different models: Model 1, which included all the identified risk factors except DPN severity, with a total of 10 variables, and Model 2, which did not include the variables that are inputs for the fuzzy system (MNSI score, vibration, and tactile sensitivities) but did include the DPN severity, for a total of eight variables.

Model 1 showed similar results for both the forward and backward stepwise regression and identified age, MNSI score, and balance problems as fall predictors (Table 3). The Hosmer–Lemeshow test indicated that the occurrence of fallers was not significantly different from the model prediction (X²=9.272; df=8; *P*=0.320), and therefore, the overall model fit was good. The Nagelkerke R² value was 0.237. The second model showed for the forward stepwise regression (Model 2a) that age, balance problems, and DPN severity were fall predictors (Table 3). The model's overall fit was good (Hosmer–Lemeshow test: X²=3.267; df=8; *P*=0.916) and presented a Nagelkerke R² value of 0.203. The backward stepwise regression of the second model (Model 2b) identified age, balance problems, sex, and DPN severity as fall predictors, with visual deficiency figuring as a covariate (Hosmer–Lemeshow test: X²=13.403; df=8; *P*=0.099). The Nagelkerke R² value was 0.247 (Table 3).

The ROC curve showed that all three models had similar performance, with Model 1 presenting the best discrimination between non-fallers and fallers and Model 2b showing a slightly better performance than Model 2a (Table 4). Since the ROC curve results were very similar for the three models, the validation of Models 1, 2a, and 2b was performed with the test sample (Table 5).

**Table 3. Final logistic regression models (n = 167).**

| Predictor | β | OR (95% CI) | P | β | OR (95% CI) | P | β | OR (95% CI) | P |
|---|---|---|---|---|---|---|---|---|---|
| Age | 0.056 | 1.06 (1.02-1.10) | 0.002* | 0.040 | 1.04 (1.01-1.07) | 0.015* | 0.040 | 1.04 (1.01-1.08) | 0.019* |
| MNSI score | 0.209 | 1.23 (1.08-1.40) | 0.001* | --- | --- | --- | --- | --- | --- |
| Self-reported balance problems | 0.976 | 2.65 (1.29-5.45) | 0.008* | 1.190 | 3.29 (1.64-6.59) | 0.001* | 1.041 | 2.83 (1.38-5.79) | 0.004* |
| DPN severity | --- | --- | --- | 0.162 | 1.18 (1.03-1.34) | 0.018* | 0.148 | 1.16 (1.01-1.33) | 0.033* |
| Sex (male) | --- | --- | --- | --- | --- | --- | −0.742 | 0.48 (0.24-0.95) | 0.035* |
| Presence of visual deficiency | --- | --- | --- | --- | --- | --- | 0.616 | 1.85 (0.90-3.82) | 0.096 |
| Constant | −4.73 | 0.009 | <0.001* | −3.50 | 0.030 | 0.001* | −3.23 | 0.040 | 0.004* |

MNSI = Michigan Neuropathy Screening Instrument; OR = odds ratio; CI = confidence interval.

**Table 4. ROC curve results for the training sample (n = 167).**

| Logistic model | AUC | 95% CI | P |
|---|---|---|---|
| Model 1 | 0.747 | 0.673-0.820 | < 0.001 |
| Model 2a | 0.728 | 0.652-0.804 | < 0.001 |
| Model 2b | 0.735 | 0.680-0.826 | < 0.001 |

AUC – area under the curve; CI = confidence interval

**Table 5. Performance measures for the test sample (n = 101).**

| Logistic model | ROC curve analysis | | | | Accuracy | Precision | Recall | Specificity | F1-Score | MCC |
|---|---|---|---|---|---|---|---|---|---|---|
| | AUC | 95% CI | P | Cut-off point | | | | | | |
| Model 1 | 0.762 | 0.655-0.869 | 0.001 | 0.14 | 0.604 | 0.239 | 0.688 | 0.588 | 0.365 | 0.202 |
| Model 2a | 0.750 | 0.628-0.872 | 0.002 | 0.13 | 0.624 | 0.238 | 0.625 | 0.624 | 0.345 | 0.184 |
| Model 2b | 0.780 | 0.679-0.881 | < 0.001 | 0.14 | 0.485 | 0.190 | 0.688 | 0.448 | 0.297 | 0.099 |

AUC – area under the curve; CI = confidence interval; MCC – Mathew's correlation analysis

The validation of the model with the test sample showed that Model 1 presented slightly better performance parameters, although all three models were valid. Model 1 achieved 60.4% accuracy, detecting 68.8% of the fallers (sensitivity or recall rate) and correctly classifying 58.8% of the non-fallers (specificity). However, there was a high proportion of false positives, leading to a precision rate of 23.9%. The F1-score, which is the harmonic mean between precision and recall, was 0.365 because of the low precision in the validation with the test set. Similarly, the high rate of false positives was responsible for an MCC of only 0.202. Still, the area under the ROC curve for the model generated by the test sample was 0.76, indicating that the discrimination of the model was fair.

With the use of the significant predictors shown in the logistic regression, the prediction for Model 1 can be expressed as follows:

$$Z = -4.73 + 0.056 \ x \ (age) + 0.209 \ x \ (MNSI \ score) + 0.976 \ x \ (self-reported \ balance \ problem) \quad (1)$$

This equation suggests that individuals with DM, older age, higher MNSI scores, and self-reported balance problems are at a greater risk of becoming recurrent fallers. Models 2a and 2b can be used to predict the probability of falls in a similar way.

## Discussion

This study aimed to identify the predictive risk factors for falls within community-dwelling older adults with DM and DPN taking into account the DPN severity. This study provides a comprehensive logistic model that allows for the estimation of the probability of a fall occurring in a real-world clinical scenario, providing a predictive model for falls in this population. The main findings showed that self-reported balance problems, DPN symptoms, and age represent the strongest risk for falls in people with DM and DPN. Additionally, DPN severity scored by the fuzzy system was also considered a predictive risk factor, suggesting that the combination of loss of sensitivity and DPN symptoms can be associated with an increased risk of falls in persons with DPN.

All three predictive models presented fair performance based on modest AUC (Area Under the Curve) values from the ROC analysis (between 0.75 and 0.78). Although Model 2b presented the higher AUC value, it also had the lowest MCC value and probably is not the best predictive model between the three developed ones. The higher the MCC score, the better the fit of the predictive model for all four possible predictive categories, i.e., true positives, false negatives, true negatives, and false positives [35]. Model 1 achieved the highest MCC; a high recall rate or sensitivity (68.8%), which demonstrates a good ability to detect fallers; and a specificity around 60%, which suggests a satisfactory rate of true negatives (i.e., non-fallers) classified correctly.

Based on the OR from Model 1, self-reported balance problems had the highest influence in falls occurrence, followed by DPN symptoms, and then age, which showed the lowest contribution. This is an interesting result since it shows that two modifiable variables (self-reported balance problems and DPN symptoms) surpass the influence of age, a non-modifiable variable, showing the potential benefit of screening and rehabilitation strategies aimed at detecting and improving balance and DPN symptoms [36] to reduce the risk of falls. Specific therapeutic exercises might improve the foot health and balance of people with DM and should be included as a complementary intervention. A recent meta-analysis showed that foot-ankle exercise programs with a duration of 8–12 weeks may result in improvements in DPN signs and symptoms [37]. Sensorimotor training also has significant effects on balance control and DPN sensory and motor signs and symptoms [38]. One study found that plantar electrical stimulation was effective for improving balance and vibratory plantar thresholds [39]. Furthermore, whole-body vibration protocols associated with balance exercises and Tai-Chi exercises have also proved to be effective in improving DPN symptoms and balance [40].

Aging has already been cited as a risk factor for falls, even for individuals without DM, due to overall frailty and the presence of comorbidities [41]. Its presence in our model supports this evidence. What the current predictive model adds to the understanding of the risk of falls is an estimate of how much this risk increases in the presence of DPN symptoms and/or balance problems. A 60-year-old individual with DM but no DPN symptoms or balance problems would have a 20.3% risk of recurrent falls. If this individual presents with an MNSI score of 4, this risk would increase 16.7%, and if they reported balance problems, the risk would increase an additional 23.9% to 60.9%. With an additional increase in the MNSI score of two points (MNSI score of 6), the probabilities would increase by an additional 10% (S2 and S3 Tables).

Although Model 2a showed slightly lower performance than Model 1, it indicated that DPN severity is also an important predictive variable for risk of falling in people with DM, suggesting that the combination of loss of sensitivity and DPN symptoms should be taken into consideration in clinical practice. Vibration perception did not represent a higher risk for falls individually, but the fewer number of subjects that had already had a fall episode may have influenced this result. Maybe if the sample size of this group was larger, this variable would clearly represent an important risk. Previous studies have shown that this variable is more efficient in discriminating early stages of DPN than tactile sensitivity assessed with a 10-g monofilament [42].

Interestingly, only a few of the traditional risk factors for falls were confirmed as predictive variables for recurrent falls in this study, with, for instance, polypharmacy and retinopathy not showing as such. Likely, the statistical power of DM- and DPN-related variables was stronger than the more general variables. This highlights that, even for the individuals who are not at a high risk for falls, it is crucial to assess the combination of DPN signs and symptoms, such as plantar sensitivity, since they are even more relevant than the traditional physical impairments associated with aging.

The strengths of this study are the sample size, the robustness of the statistical modeling procedure, and the population studied (community-dwelling persons who are usually less physically impaired than hospital-based populations). In addition, this is the first study to employ a modeling analysis to develop a predictive tool for falls in individuals with DM and DPN using clinical variables from commonly used instruments by healthcare professionals. This tool could benefit healthcare professionals in screening community-dwelling older individuals with various levels of fall risk. Our limitations include the uneven group sizes, although the statistical model is specific for this design. Because the assessment took place in a population campaign, the questionnaires and falls incidence were self-reported, and objective scales were not used. This fact can introduce imprecision on a certain level; however, the sample size was big enough to represent results that are statistically valid.

In conclusion, this study shows that self-reported balance problems, DPN symptoms, severity, and age are fall risk factors in community-dwelling individuals with DM, and these variables should be assessed and managed in clinical practice to recognize individuals at risk and prevent recurrent falls in this population. Finally, this study shows the possibility of predicting the risk for recurrent falls using a model based on clinical variables commonly assessed by healthcare professionals in a real-world setting.

## Supporting information

**S1 File. Equations 1 and 2 generate by the logistic regression.**
(DOCX)

**S1 Table. Simple logistic regression analyses for potential predictors of falls.**
(DOCX)

**S2 Table. Individual data for Model 1 prediction including all risk factors, except DPN severity.**
(DOCX)

**S3 Table. Individual data for Model 2A prediction including all risk factors, except fuzzy system variables (MNSI score, vibration, and tactile sensitivities).**
(DOCX)

## Acknowledgments

The authors acknowledge the Associação Nacional de Assistência ao Diabético (ANAD) for providing the subjects for this research.

## Author contributions

**Conceptualization:** Eneida Yuri Suda, Cristina Dallemole Sartor, Isabel C.N. Sacco.

**Data curation:** Eneida Yuri Suda, Cristina Dallemole Sartor, Isabel C.N. Sacco.

**Formal analysis:** Eneida Yuri Suda, Cristina Dallemole Sartor, Ricky Watari, Isabel C.N. Sacco.

**Funding acquisition:** Isabel C.N. Sacco.

**Investigation:** Cristina Dallemole Sartor, Eunice Young Docko, Isabel C.N. Sacco.

**Methodology:** Eneida Yuri Suda, Cristina Dallemole Sartor, Isabel C.N. Sacco.

**Project administration:** Cristina Dallemole Sartor, Isabel C.N. Sacco.

**Resources:** Cristina Dallemole Sartor, Isabel C.N. Sacco.

**Software:** Cristina Dallemole Sartor, Isabel C.N. Sacco.

**Supervision:** Cristina Dallemole Sartor, Isabel C.N. Sacco.

**Validation:** Eneida Yuri Suda, Cristina Dallemole Sartor, Isabel C.N. Sacco.

**Visualization:** Cristina Dallemole Sartor, Isabel C.N. Sacco.

**Writing – original draft:** Eneida Yuri Suda, Cristina Dallemole Sartor, Ricky Watari, Isabel C.N. Sacco.

**Writing – review & editing:** Eneida Yuri Suda, Cristina Dallemole Sartor, Anice de Campos Pássaro, Ricky Watari, Eunice Young Docko, Isabel C.N. Sacco.

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
