## [Decision Letter · Decision Letter 0]

29 Oct 2025

Dear Dr. Sacco,

Thank you for submitting your manuscript to PLOS ONE. After careful consideration, we feel that it has merit but does not fully meet PLOS ONE’s publication criteria as it currently stands. Therefore, we invite you to submit a revised version of the manuscript that addresses the points raised during the review process.

Please address all the reviewers’ concerns thoroughly This manuscript requires a minor revision Carefully review and follow all the journal’s guidelines 

We look forward to receiving your revised manuscript.

Kind regards,

Fredirick Lazaro mashili, MD, PhD

Academic Editor

PLOS ONE

Journal Requirements:

“This work was supported by the National Council for Scientific and Technological Development, Brazil (CNPq) [28/2018 FOCA Trial 407252/2018-5]. Sacco is a fellow level 1B of the National Council for Scientific and Technological Development (CNPq), Brazil (Process: 302558/2022-5) and Docko holded a scholarship from CNPq (PIBIC). The funders do not have any role in the study and do not have any authority over any study activity or in the decision to submit the report for publication.”

3. Thank you for uploading your study's underlying data set. Unfortunately, the repository you have noted in your Data Availability statement does not qualify as an acceptable data repository according to PLOS's standards.

“This work was supported by the National Council for Scientific and Technological Development, Brazil (CNPq) [28/2018 FOCA Trial 407252/2018-5]. Sacco is a fellow level 1B of the National Council for Scientific and Technological Development (CNPq), Brazil (Process: 302558/2022-5) and Docko holded a scholarship from CNPq (PIBIC). The funders do not have any role in the study and do not have any authority over any study activity or in the decision to submit the report for publication.”

“This work was supported by the National Council for Scientific and Technological Development, Brazil (CNPq) [28/2018 FOCA Trial 407252/2018-5]. Sacco is a fellow level 1B of the National Council for Scientific and Technological Development (CNPq), Brazil (Process: 302558/2022-5) and Docko holded a scholarship from CNPq (PIBIC). The funders do not have any role in the study and do not have any authority over any study activity or in the decision to submit the report for publication.”

Additional Editor Comments (if provided):

Please address all the comments made by both the reviewers. Please make sure to review and abide to all journal’s formatting and specific requirements

Reviewers' comments:

Reviewer's Responses to Questions

**Comments to the Author**

1. Is the manuscript technically sound, and do the data support the conclusions?

Reviewer #1: Yes

Reviewer #2: Yes

2. Has the statistical analysis been performed appropriately and rigorously?

Reviewer #1: Yes

Reviewer #2: Yes

3. Have the authors made all data underlying the findings in their manuscript fully available?

Reviewer #1: Yes

Reviewer #2: Yes

4. Is the manuscript presented in an intelligible fashion and written in standard English?

Reviewer #1: Yes

Reviewer #2: Yes

Reviewer #1: The authors used a logistic model to determine the interplay between diabetes related risks with risks of fall in a community dwelling adults with DPN. They used a robust methodology to develop and test/validate the model. The sample size is reasonable given the number of predictors they tested.

The study is meticulously thought of, designed and the manuscript very well written. Despite these strengths, the title and main aim are somehow NOT CLEAR. While the authors developed and validated the model, the title sounds as if they were merely determining associations. In explanatory logistic regression models the emphasis is mainly on adjusted ORs and p-values (associations). In this case splitting data into training and testing is not necessary. In prediction models however like what the authors have done, AUC, sensitivity and calibrations of the model are the main emphasis.

It would be great if the authors consider revising their title and main aim (also enriching their discussion) to reflect exactly what they have done.

Reviewer #2: This paper is well written and scientifically sound, with the method section well elaborated, ensuring reproducibility. The scientific findings are backed up and supported with relevant literature. However, the authors are advised to follow the journal instruction on formatting and referencing styles.

**Do you want your identity to be public for this peer review?** For information about this choice, including consent withdrawal, please see our Privacy Policy

Reviewer #1: **Yes: ** Fredirick mashili

Reviewer #2: **Yes: ** George Gabriel Mkumbi

---

## [Author Response · Author response to Decision Letter 1]

10 Nov 2025

Response to reviewers

São Paulo, November 5th 2025

Dear Dr Fredirick Lazaro Mashili,

Academic Editor

PLOS ONE

We would like to thank for the opportunity of submitting a revised version of the manuscript “Interplay between diabetes-related risk factors for falls in community-dwelling people with diabetic peripheral neuropathy based on a logistic regression model” (PONE-D-24-41153). This document contains the responses to all comments raised by the academic editor and reviewers on a point-by-point basis. We hope our task fulfils the reviewers’ and editors’ expectations. Thank you once again for the opportunity, and please do not hesitate to contact us if you have any further queries and requests.

Yours sincerely. On behalf of all authors,

PhD. Professor Isabel de Camargo Neves Sacco

Associate Professor - MS5 III

Head of the Laboratory of Biomechanics of Human Movement and Posture

Physical Therapy, Speech and Occupational Therapy department – School of Medicine – University of São Paulo

RESPONSES TO THE ACADEMIC EDITOR AND REVIEWERS' COMMENTS ON A POINT-BY-POINT BASIS

Journal Requirements:

Authors’ response: Thank you for pointing out. We carefully checked the requirements and made the necessary amendments (headings, spacing, equation, tables citation and captions). We also changed the first page (title, authors, affiliations) according to the formatting guideline. The changes that corresponded to style adjustments were not marked.

“This work was supported by the National Council for Scientific and Technological Development, Brazil (CNPq) [28/2018 FOCA Trial 407252/2018-5]. Sacco is a fellow level 1B of the National Council for Scientific and Technological Development (CNPq), Brazil (Process: 302558/2022-5) and Docko holded a scholarship from CNPq (PIBIC). The funders do not have any role in the study and do not have any authority over any study activity or in the decision to submit the report for publication.”

Authors’ response: The funding statement was amended according to the recommendation:

“This work was supported by the National Council for Scientific and Technological Development, Brazil (CNPq) [28/2018 FOCA Trial 407252/2018-5]. Sacco is a fellow level 1B of the National Council for Scientific and Technological Development (CNPq), Brazil (Process: 302558/2022-5) and Docko holded a scholarship from CNPq (PIBIC). All funders did not have any role in the study and do not have any role in the study and do not have any authority over any study activity or in the decision to submit the report for publication. There was no additional external funding received for this study.”

3. Thank you for uploading your study's underlying data set. Unfortunately, the repository you have noted in your Data Availability statement does not qualify as an acceptable data repository according to PLOS's standards.

Authors’ response: The data set was uploaded at figshare as requested. The datasets generated and/or analyzed during the current study are accessible as anonymized data at https://doi.org/10.6084/m9.figshare.30546620.

“This work was supported by the National Council for Scientific and Technological Development, Brazil (CNPq) [28/2018 FOCA Trial 407252/2018-5]. Sacco is a fellow level 1B of the National Council for Scientific and Technological Development (CNPq), Brazil (Process: 302558/2022-5) and Docko holded a scholarship from CNPq (PIBIC). The funders do not have any role in the study and do not have any authority over any study activity or in the decision to submit the report for publication.”

“This work was supported by the National Council for Scientific and Technological Development, Brazil (CNPq) [28/2018 FOCA Trial 407252/2018-5]. Sacco is a fellow level 1B of the National Council for Scientific and Technological Development (CNPq), Brazil (Process: 302558/2022-5) and Docko holded a scholarship from CNPq (PIBIC). The funders do not have any role in the study and do not have any authority over any study activity or in the decision to submit the report for publication.”

Authors’ response: The funding statement was amended as “This work was supported by the National Council for Scientific and Technological Development, Brazil (CNPq) [28/2018 FOCA Trial 407252/2018-5]. Sacco is a fellow level 1B of the National Council for Scientific and Technological Development (CNPq), Brazil (Process: 302558/2022-5) and Docko holded a scholarship from CNPq (PIBIC). All funders did not have any role in the study and do not have any role in the study and do not have any authority over any study activity or in the decision to submit the report for publication. There was no additional external funding received for this study.”

The acknowledgment section was also updated as follows:

“The authors acknowledge the Associação Nacional de Assistência ao Diabético (ANAD) for providing the subjects for this research.”

Authors’ response: Thank you for pointing out, but no recommendations were made by the reviewers.

Authors’ response: The reference list was reviewed, and no changes were necessary.

Additional Editor Comments (if provided):

Please address all the comments made by both the reviewers. Please make sure to review and abide to all journal’s formatting and specific requirements.

Authors’ response: The reviewers’ comments are addressed below on a point-by point basis.

Reviewers' comments:

Reviewer #1: The authors used a logistic model to determine the interplay between diabetes related risks with risks of fall in a community dwelling adults with DPN. They used a robust methodology to develop and test/validate the model. The sample size is reasonable given the number of predictors they tested. The study is meticulously thought of, designed and the manuscript very well written. Despite these strengths, the title and main aim are somehow NOT CLEAR. While the authors developed and validated the model, the title sounds as if they were merely determining associations. In explanatory logistic regression models the emphasis is mainly on adjusted ORs and p-values (associations). In this case splitting data into training and testing is not necessary. In prediction models however like what the authors have done, AUC, sensitivity and calibrations of the model are the main emphasis. It would be great if the authors consider revising their title and main aim (also enriching their discussion) to reflect exactly what they have done.

Authors’ response: Thank you very much for your kind consideration and suggestions. We made some adjustments according to the reviewer’s points:

Title – “Predictive effects of diabetes-related risk factors for falls in community-dwelling people with diabetic peripheral neuropathy based on a logistic regression model”

Aim – We believe that the aim in the manuscript is already bringing emphasis to the development of a predictive model, but we did a slight change to enhance it, both in the main text as in the abstract:

• Abstract: “This study aimed to identify the predictive effects of different aspects of diabetic peripheral neuropathy (DPN) and other already known risk factors for falls through a comprehensive logistic model within community-dwelling older adults with diabetes and DPN. This paper also provides a model that estimates the probability of a fall occurring in a real-world clinical scenario.”

• Main text: “Thus, the aim of this study was to identify predictive risk factors for falls within community-dwelling older adults with DM and DPN, taking into account the DPN severity degree, and to provide a predictive comprehensive logistic model that allows for the estimation of the probability of a fall occurring in a real-world clinical scenario.”

Discussion – “This study provides a comprehensive logistic model that allows for the estimation of the probability of a fall occurring in a real-world clinical scenario, providing a predictive model for falls in this population.”

Overall, we believe that the discussion is already based in the presentation of a predictive model, as suggested by the reviewer, so we did not make any additional amendments. For instance, in the second paragraph we discuss the performance of all the predictive models presented. Here are some additional examples:

“The main findings showed that self-reported balance problems, DPN symptoms, and age represent the strongest risk for falls in people with DM and DPN. Additionally, DPN severity scored by the fuzzy system was also considered a predictive risk factor, suggesting that the combination of loss of sensitivity and DPN symptoms can be associated with an increased risk of falls in persons with DPN."

“Aging has already been cited as a risk factor for falls, even for individuals without DM, due to overall frailty and the presence of comorbidities (James et al., 2020). Its presence in our model supports this evidence. What the current predictive model adds to the understanding of the risk of falls is an estimate of how much this risk increases in the presence of DPN symptoms and/or balance problems.”

“Although Model 2a showed slightly lower performance than Model 1, it indicated that DPN severity is also an important predictive variable for risk of falling in people with DM…”

“In addition, this is the first study to employ a modeling analysis to develop a predictive tool for falls in individuals with DM and DPN using clinical variables from commonly used instruments by healthcare professionals.”

“Finally, this study shows the possibility of predicting the risk for recurrent falls using a model based on clinical variables commonly assessed by healthcare professionals in a real-world setting.”

Reviewer #2:

This paper is well written and scientifically sound, with the method section well elaborated, ensuring reproducibility. The scientific findings are backed up and supported with relevant literature. However, the authors are advised to follow the journal instruction on formatting and referencing styles.

Authors’ response: Thank you very much for this comment, we are pleased that we met the reviewer’s expectations. The formatting and reference styles were checked and revised, as requested.

---

## [Decision Letter · Decision Letter 1]

18 Dec 2025

Predictive effects of diabetes-related risk factors for falls in community-dwelling people with diabetic peripheral neuropathy based on a logistic regression model

PONE-D-24-41153R1

Dear Dr. Sacco,

We’re pleased to inform you that your manuscript has been judged scientifically suitable for publication and will be formally accepted for publication once it meets all outstanding technical requirements.

Kind regards,

Fredirick Lazaro mashili, MD, PhD

Academic Editor

PLOS One

Additional Editor Comments (optional):

All previously raised concerns have been thoroughly addressed

Reviewers' comments:

Reviewer's Responses to Questions

**Comments to the Author**

Reviewer #1: All comments have been addressed

Reviewer #2: All comments have been addressed

2. Is the manuscript technically sound, and do the data support the conclusions?

Reviewer #1: Yes

Reviewer #2: Yes

3. Has the statistical analysis been performed appropriately and rigorously?

Reviewer #1: Yes

Reviewer #2: Yes

4. Have the authors made all data underlying the findings in their manuscript fully available?

Reviewer #1: Yes

Reviewer #2: Yes

5. Is the manuscript presented in an intelligible fashion and written in standard English?

Reviewer #1: Yes

Reviewer #2: Yes

Reviewer #1: The authors have sufficiently addressed all the previous comments raised by the reviewers. The manuscript is now good for publication in PLOS ONE.

Reviewer #2: I would like to acknowledge the effort that the authors have put into production of this paper. After reviewing all the responses from all the issues raised by reviewers, i can confirm that all were sufficiently addressed and therefore this paper is deemed worth for publication.

**Do you want your identity to be public for this peer review?** For information about this choice, including consent withdrawal, please see our Privacy Policy

Reviewer #1: **Yes: ** Fredirick mashili

Reviewer #2: **Yes: ** George Gabriel Mkumbi

---

## [Editor Report · Acceptance letter]

PONE-D-24-41153R1

PLOS One

Dear Dr. Sacco,

I'm pleased to inform you that your manuscript has been deemed suitable for publication in PLOS One. Congratulations! Your manuscript is now being handed over to our production team.

Kind regards,

on behalf of

Dr. Fredirick Lazaro mashili

Academic Editor

PLOS One